# Calcium, Phosphorus and Magnesium Abnormalities Associated with COVID-19 Infection, and Beyond

**DOI:** 10.3390/biomedicines11092362

**Published:** 2023-08-23

**Authors:** Lucreția Anghel, Corina Manole, Aurel Nechita, Alin Laurențiu Tatu, Bogdan Ioan Ștefănescu, Luiza Nechita, Camelia Bușilă, Pușica Zainea, Liliana Baroiu, Carmina Liana Mușat

**Affiliations:** 1Clinical Medical Department, Faculty of Medicine and Pharmacy, ‘Dunarea de Jos’ University, 800008 Galati, Romania; anghel_lucretia@yahoo.com (L.A.); nechitaaurel@yahoo.com (A.N.); dralin_tatu@yahoo.com (A.L.T.); nechitaluiza2012@yahoo.com (L.N.); camelia_busila@yahoo.com (C.B.); lilibaroiu@yahoo.com (L.B.); 2‘Sf. Apostol Andrei’ Clinical Emergency County Hospital, 800578 Galati, Romania; b_stefanescu@yahoo.com (B.I.Ș.); carmina.musat@ugal.ro (C.L.M.); 3‘Sf. Ioan’ Clinical Hospital for Children, 800487 Galati, Romania; 4‘Sf. Cuv. Parascheva’ Clinical Hospital of Infectious Diseases, 800179 Galati, Romania; 5Multidisciplinary Integrated Center of Dermatological Interface Research MIC-DIR, 800010 Galati, Romania; 6Clinical Surgery Department, Faculty of Medicine and Pharmacy, ‘Dunarea de Jos’ University, 800008 Galati, Romania; 7Clinical Emergency County Hospital Braila, 810318 Braila, Romania; pusica.pusica1967@gmail.com; 8Department of Morphological and Functional Sciences, Faculty of Medicine and Pharmacy, ‘Dunarea de Jos’ University, 800008 Galati, Romania

**Keywords:** COVID-19, phosphocalcium, hypoparathyroidism, parathyroid hormone, post-COVID complications, hypocalcemia, hypomagnesemia, hypophosphatemia

## Abstract

The coronavirus disease (COVID-19) pandemic caused by the novel coronavirus SARS-CoV-2 has had a profound impact on global health, leading to a surge in research to better understand the pathophysiology of the disease. Among the various aspects under investigation, disruptions in mineral homeostasis have emerged as a critical area of interest. This review aims to provide an overview of the current evidence linking calcium, phosphorus and magnesium abnormalities with COVID-19 infection and explores the potential implications beyond the acute phase of the disease. Beyond the acute phase of COVID-19, evidence suggests a potential impact of these mineral abnormalities on long-term health outcomes. Persistent alterations in calcium, phosphorus and magnesium levels have been linked to increased cardiovascular risk, skeletal complications and metabolic disorders, warranting continuous monitoring and management in post-COVID-19 patients.

## 1. Introduction

The COVID-19 pandemic has had significant global health implications, affecting millions of individuals worldwide. While the acute respiratory symptoms of COVID-19 have been extensively studied, emerging evidence suggests that the disease can have systemic effects, including disturbances in phosphocalcium metabolism [1]. This manuscript review aims to summarize and critically analyze the current research on phosphocalcium metabolism disorders post-COVID, providing insights into the potential mechanisms and clinical implications [2,3,4].

Phosphocalcium metabolism plays a crucial role in maintaining the balance of phosphorus and calcium in the body, which is essential for various physiological processes, including bone health, muscle function, nerve transmission and cellular signaling [5]. Any disturbances in this delicate equilibrium can lead to a spectrum of metabolic disorders, with significant implications for an individual’s overall well-being.

This manuscript review aims to provide a comprehensive analysis of the current state of research on phosphocalcium metabolism disorders post-COVID. It will delve into the existing literature to explore the impact of COVID-19 on phosphorus, calcium and parathyroid hormone levels, along with the prevalence and clinical manifestations of these disturbances in recovering individuals [6].

Understanding the consequences of COVID-19 on phosphocalcium metabolism is of paramount importance for healthcare professionals managing post-COVID patients. By identifying and addressing these metabolic disturbances, healthcare providers can potentially mitigate long-term complications and improve patient outcomes. Furthermore, this review will highlight potential avenues for future research to enhance our understanding of the pathophysiological mechanisms involved and develop effective strategies for managing phosphocalcium metabolism disorders in the aftermath of COVID-19.

In the following sections, this manuscript review will present a comprehensive overview of the research findings, providing valuable insights into the clinical implications and potential interventions for individuals experiencing phosphocalcium metabolism disorders after recovering from COVID-19.

## 2. Physiology

### 2.1. Phosphocalcium Metabolism

The minerals calcium and phosphorus are essential components in the process of bone mineralization. Both minerals are found in bones and can be dissolved in serum or found intracellularly in soft tissues. Calcium and phosphorus are stored within the bone structure in the form of a crystalline compound known as hydroxyapatite. This compound not only acts as a storage site for these minerals but also serves as the fundamental building block of the bone, providing the necessary strength and support to bear the weight of the body. Calcium and phosphorus in serum exist in three forms: ionized, bound to albumin or present in other ion complexes [2]. Calcium is involved in various physiological processes, including muscle contraction, hormone and neurotransmitter release, as well as enzyme activation and coagulation pathways [3]. The presence of soluble calcium and phosphorus in serum is crucial for maintaining homeostasis and facilitating the proper functioning of various systems, including the nervous and muscular systems. Consequently, these levels are tightly regulated by hormones through processes such as intestinal absorption, bone resorption, renal excretion and reabsorption [7].

The regulation and monitoring of calcium levels in the bloodstream are closely managed through the utilization of calcium-sensing receptors (CaSR) [8]. Calcium-sensing receptors (CaSRs) are a type of G-protein coupled receptor that is predominantly located in the kidneys and parathyroid glands. When calcium levels are high, an excessive amount of calcium binds to calcium-sensing receptors (CaSRs), leading to a decrease in the synthesis and secretion of parathyroid hormone. Additionally, it causes a reduction in the reabsorption of calcium by the kidneys [8].

Phosphorus serves as a structural constituent of bone in the form of hydroxyapatite. Phosphorus primarily serves the crucial role of facilitating cellular energy production and supporting metabolic processes by acting as a constituent of adenosine triphosphate (ATP). The maintenance of serum phosphorus homeostasis is achieved through the ongoing processes of bone mineralization and resorption, which are regulated by a delicate equilibrium between osteoblasts, the cells responsible for bone formation, and osteoclasts, the cells involved in bone reabsorption [2,4].

The regulation of calcium and phosphorus in the body is primarily controlled by three hormones: vitamin D 25(OH), parathyroid hormone (PTH), and fibroblast growth factor 23 (FGF23) [3].

### 2.2. Regulation by Vitamin D

Vitamin D2 and D3 are converted to an active form known as vitamin D1,25(OH)2 or calcitriol (Figure 1). The synthesis of vitamin D3 (cholecalciferol) in the skin of animals begins with the conversion of 7-dehydrocholesterol through exposure to UVB radiation and heat. Vitamin D2, also known as ergocalciferol, is derived from plant and fungal sources [2]. Additional sources of vitamin D include fish liver, oily fish and supplementation in milk and orange juice. Within the hepatic system, the hydroxylation process takes place, wherein D2 and D3 undergo conversion into calcidiol, also known as vitamin D25(OH) [2]. The primary role of this intermediate metabolite is to serve as the reservoir for vitamin D. Within the renal system, calcidiol undergoes additional hydroxylation through the enzymatic action of 1α-hydroxylase, leading to the production of calcitriol, an active form of vitamin D. This process ultimately enhances the absorption of calcium and phosphorus in the intestines [1,2,4].

### 2.3. Regulation by PTH

Parathyroid hormone (PTH), which is secreted by the four parathyroid glands, plays a crucial role in regulating calcium and phosphorus levels in bone, the gastrointestinal tract and the kidneys. This schema is summarized in Figure 2. Within the renal tubules, parathyroid hormone (PTH) has the effect of augmenting the reabsorption of calcium while simultaneously enhancing the excretion of phosphorus [2]. Within the gastrointestinal tract, parathyroid hormone (PTH) facilitates the process of reabsorption of calcium and phosphorus. This is achieved indirectly through the stimulation of 1α-hydroxylase, as supported by previous studies [3]. Within the skeletal system, parathyroid hormone (PTH) serves as a stimulant for both osteoblasts and osteoclasts, leading to an elevation in bone turnover and ultimately resulting in bone resorption. The process of bone resorption leads to elevated levels of phosphorus and calcium. However, it is important to note that parathyroid hormone (PTH) also enhances the excretion of phosphorus in the kidneys [9]. Consequently, the overall outcome is an elevation in serum calcium levels and a reduction in serum phosphorus levels [2,3]. The impact of parathyroid hormone (PTH) on renal function is characterized by a rapid onset, manifesting within minutes.

### 2.4. Regulation by FGF23

FGF23 is a bone- and bone-marrow-derived hormone involved in the regulation of phosphorus levels. It functions as a phosphatonin, playing a crucial role in promoting the excretion of phosphorus in urine [3] (Figure 3). Circulating FGF23 activates FGF-R1 tyrosine kinase receptors together with the co-receptor alpha KLOTHO [3]. FGF23 inhibits 1α-hydroxylase and lowers calcitriol levels [3,5]. High blood phosphate levels cause the skeletal system to release FGF23, which inhibits phosphate reabsorption in the renal tubules and calcitriol production. FGF23 also promotes the production of 24α-hydroxylase, an enzyme responsible for the conversion of vitamin D25(OH) to vitamin D24,25(OH)2, which is an inactive metabolite.

Emerging research suggests a potential association between fibroblast growth factor 23 (FGF23), phosphocalcium metabolism and COVID-19 [10,11]. FGF23, a hormone primarily associated with regulating phosphate and vitamin D levels in the body, has gained attention as a potential biomarker in the context of COVID-19 due to its role in maintaining mineral homeostasis and its interactions with the immune system [12].

Several studies have explored this correlation, shedding light on the possible interplay between FGF23, phosphocalcium metabolism and COVID-19 severity:

Research has shown that FGF23 levels tend to be elevated in severe COVID-19 cases. Higher FGF23 levels have been associated with worse clinical outcomes, including increased mortality rates and respiratory distress [13]. This suggests a potential link between FGF23 dysregulation, disrupted phosphocalcium metabolism and the severity of COVID-19.

FGF23 and vitamin D are intricately connected in regulating phosphocalcium homeostasis. Certain studies have reported an inverse relationship between FGF23 and vitamin D levels in COVID-19 patients [14,15,16]. This could imply that the disruption of the FGF23–vitamin D axis might contribute to the dysregulation of phosphocalcium metabolism observed in severe cases.

The exact mechanisms underlying the association between FGF23, phosphocalcium metabolism and COVID-19 remain an active area of investigation. It is speculated that the inflammatory response triggered by COVID-19 could impact FGF23 production and function, potentially leading to disturbances in phosphocalcium balance.

While these findings suggest a potential association between FGF23, phosphocalcium metabolism and COVID-19, it is important to note that the field of COVID-19 research is still evolving, and more studies are needed to establish definitive causal relationships. Further research is required to determine whether FGF23 dysregulation plays a causal role in the severity of COVID-19 or if it serves as an indicator of disease progression.

Understanding the intricate relationship between FGF23, phosphocalcium metabolism and COVID-19 could potentially open avenues for novel therapeutic interventions or prognostic indicators. As research continues to unfold, it will be essential to conduct well-designed studies to elucidate the precise mechanisms and clinical implications of this association.

## 3. Methods

### Literature Search Strategy

A comprehensive search strategy was developed to identify relevant articles from electronic databases such as PubMed, Google Scholar, or Scopus which were searched using keywords including “calcium metabolism disorders in COVID-19”, “phosphocalcium metabolism”, “hypophosphatemia in COVID-19”, “hypoparathyroidism in COVID-19”, “hypocalcemia in COVID-19” and “post-COVID complications”. The search was limited to articles from the last 4 years to ensure the inclusion of recent research.

## 4. Incidence of Phosphocalcium Metabolism Disorders Related to COVID-19

The literature search revealed 240 studies, 31 of which were ultimately included in this review: 7 articles about hypocalcemia, 12 articles about hypophosphatemia and 12 articles about hypovitaminosis D.

### 4.1. Hypocalcemia

Low calcium levels were a common biochemical result during the first pandemic outbreak in Europe. Beginning with the seminal observation of a case of severe acute hypocalcemia in a patient previously thyroidectomized with infection with SARS-CoV-2 [17], several studies undertaken worldwide to evaluate the possible causal or causal relationship between COVID-19 and hypocalcemia revealed an unexpectedly high prevalence of low calcium levels, ranging from 62.6 to 87.2% of patients, depending on the hypocalcemia definition used [5,18,19].

In their study, Cappellini et al. (letter to the editor) observed a significant decrease in both serum total calcium and whole blood actual ionized calcium in COVID-19 patients. This observation highlights disturbances in calcium levels in individuals affected by the virus.

In a recent study, researchers examined the relationship between calcium levels and various parameters related to inflammation, coagulopathy and organ injury in individuals with COVID-19 [20]. The findings revealed a negative association between calcium levels and these parameters, suggesting that higher calcium levels were linked to better outcomes. Additionally, the study identified hypocalcemia, or low calcium levels, as an independent risk factor for adverse effects in COVID-19 patients [21]. Reduced total calcium levels in COVID-19 patients can sometimes be attributed to low serum albumin levels [22]. COVID-19 is known to cause various systemic effects, including changes in blood chemistry and electrolyte balance. One such effect is hypoalbuminemia, where COVID-19 patients may experience lower levels of serum albumen in their blood [23].

Serum albumin plays a crucial role in transporting calcium in the bloodstream. When serum albumin levels are low, there is a reduction in the binding capacity for calcium, leading to a decrease in measured total calcium levels. This can give a misleading impression that the patient is experiencing hypocalcemia (low calcium levels).

In reality, 50% of calcium in the blood is in its ionized form, which is the biologically active and physiologically relevant form. The level of ionized calcium remains relatively stable, even when total calcium levels appear reduced due to low albumin levels.

To accurately assess the calcium status of COVID-19 patients with hypoalbuminemia, clinicians may consider calculating the corrected calcium levels by adjusting for serum albumin concentrations. In clinical practice, ionized calcium can be measured directly in addition to total calcium levels. Ionized calcium represents the physiologically active free form of calcium in the blood, and it is an important parameter for assessing calcium homeostasis and its impact on various physiological processes.

Total calcium measurements include both ionized calcium and calcium bound to proteins, primarily albumin. Since the level of albumin can vary, especially in critically ill patients, measuring ionized calcium provides a more accurate reflection of the biologically active calcium concentration. This correction helps to provide a more precise reflection of the biologically active ionized calcium in the blood and assists in making appropriate clinical decisions [24].

This highlights the importance of calcium as a reliable and easily measurable biomarker for assessing disease severity in COVID-19. The results of this study provide valuable insights into the potential role of calcium in predicting and managing COVID-19 outcomes [25]. The findings of this research were supported by subsequent systematic reviews and meta-analyses, which also concluded that there is a significant correlation between hypocalcemia and various indicators of disease severity (Table 1) [26,27]. These indicators include hospitalization rates, length of hospital stay, admission to the intensive care unit and the risk of mortality [22,28].

### 4.2. Hypophosphatemia

There is a suggested association between disturbances in phosphate metabolism and the impact of COVID-19 on the homeostasis of vitamin D, calcium and phosphorus. The prevalence of vitamin D deficiency is high among individuals with severe COVID-19 infection [40]. Povaliaeva et al. recently conducted a study that established an association between abnormal kidney function, abnormal vitamin D metabolism and hypophosphatemia (Table 1) [41]. According to the authors, individuals with severe COVID-19 manifested atypical vitamin D metabolism, increased serum creatinine levels and decreased serum phosphate levels (Figure 4) [41].

Vitamin D insufficiency has been associated with reduced levels of blood calcium and phosphate as a result of renal excretion and impaired absorption in the intestines [42]. The parathyroid hormone (PTH) is known to exhibit several effects in response to low blood calcium levels. It stimulates the reabsorption of calcium in the bones, kidneys and intestines. However, it also leads to an increased loss of phosphate via the kidneys [42]. In relation to this matter, Pal et al. conducted a comparative analysis of blood phosphorus levels between COVID-19 patients and a control group of healthy individuals matched for age, sex and vitamin D status. A study conducted by researchers revealed a decrease in phosphorus levels among individuals diagnosed with COVID-19 [26]. The aforementioned discovery presents a divergence from the previously posited pathways regarding the impact of vitamin D deprivation on disruptions in phosphate metabolism during COVID-19 infection.

There is a notable prevalence of hypophosphatemia among individuals diagnosed with COVID-19 who also have end-stage renal diseases (ESRD) [43].

Mechanisms that are linked to the occurrence of hypophosphatemia in individuals with renal dysfunction have been documented in previous studies [34]. The occurrence of acute kidney injury (AKI) can be attributed to various mechanisms in the context of COVID-19. Prerenal acute kidney injury (AKI) can manifest as a result of profound dehydration and the accumulation of water within the pulmonary interstitial tissue. The predominant etiology of renal dysfunction in individuals afflicted with COVID-19 has been documented as proximal tubulopathies, which can be attributed to either vascular-related factors or direct viral infiltration. This pathophysiological process leads to the subsequent occurrence of electrolyte depletion and subsequent development of additional complications [44].

Malnutrition and the inadequate dietary intake of phosphate-rich foods can lead to hypophosphatemia in COVID-19 patients. During severe illness, especially in hospitalized patients, there might be a reduced oral intake or impaired absorption of essential nutrients, including phosphate. Additionally, COVID-19 patients with concurrent end-stage renal diseases (ESRD) may already have compromised nutrient absorption due to renal dysfunction, further exacerbating the risk of developing hypophosphatemia.

COVID-19 patients, especially those with severe respiratory distress, may experience respiratory alkalosis, a condition characterized by reduced levels of carbon dioxide (CO_2_) in the blood. This occurs when patients breathe rapidly, leading to CO_2_ elimination and a shift towards alkalinity. Respiratory alkalosis can cause phosphate to move from the extracellular fluid into the intracellular compartment, resulting in decreased serum phosphate levels.

Both poor nutrition and respiratory alkalosis can act synergistically to contribute to hypophosphatemia in COVID-19 patients with ESRD. Additionally, the combination of these factors with other mechanisms mentioned earlier, such as renal tubulopathies and abnormal vitamin D metabolism, can further compound the phosphate metabolism disturbances in these patients (Figure 5).

### 4.3. Hypovitaminosis D

The importance of the strong association between VD and COVID-19 emerged during the initial stages of the pandemic, as VD has been widely recognized for its role in modulating both innate and adaptive immune responses [45]. Vitamin D (VD) has been recognized for its antimicrobial properties and ability to inhibit viral activity. It also plays a role in regulating the adaptive immune response by promoting a transition from a pro-inflammatory state to a tolerogenic state. This results in the downregulation of immune responses mediated by T-helper-1 lymphocytes, the inhibition of pro-inflammatory cytokine production and the promotion of regulatory T-cell maturation [46].

Several studies have observed an association between low vitamin D levels and worse COVID-19 outcomes, including an increased risk of severe illness, ICU admission, and mortality [47,48]. Vitamin D has immunomodulatory effects and can influence the expression of genes involved in the immune response. Deficiencies in vitamin D may lead to dysregulated immune responses, increased inflammation and impaired lung function, all of which can contribute to disease severity in COVID-19.

On the other hand, deficiencies in calcium, phosphorus and magnesium may also impact immune function and contribute to an increased susceptibility to viral infections, including COVID-19. These cations are essential for various cellular processes, including immune cell function and cytokine signaling [42]. A deficiency in any of these cations could impair the immune response and potentially compromise the body’s ability to fight off viral infections.

COVID-19 itself can disrupt calcium, phosphorus and magnesium homeostasis through multiple mechanisms. The severe inflammatory response triggered by the virus can cause cytokine storm and lead to altered levels of these cations [49]. Additionally, factors like respiratory alkalosis, kidney dysfunction and electrolyte shifts due to fluid imbalances in critically ill patients can further contribute to cation abnormalities.

Vitamin D (VD) has been recognized for its significant involvement in various metabolic pathways related to musculoskeletal health [41]. Previous research has demonstrated that supplementation with VD has been shown to confer advantages in muscle recovery following periods of intense physical activity and tissue damage [31,50]. The studies found that VD levels were able to predict and exert an influence on the duration of illness and the time it took for recovery following an episode of acute severe pneumonia (Table 2). To this day, the role of vitamin D in the occurrence of long COVID has only been examined in a limited number of small-scale studies [51,52,53]. In a recent pilot study involving elderly patients recovering from acute COVID-19, the effectiveness of a six-week therapy with 2000 IU/day of cholecalciferol compared to placebo was examined. The study found that cholecalciferol therapy resulted in a reduction in creatinine kinase values and demonstrated a positive trend in improving overall health and physical well-being [54,55,56,57].

### 4.4. Hypoparathyroidism

The COVID-19 disease has been documented as a potential trigger for the decompensation of pre-existing primary hypoparathyroidism, which was previously well-tolerated by affected individuals. In their study, Bossoni et al. [17] documented a clinical case involving a 72-year-old female patient who had previously undergone thyroidectomy [72]. The patient exhibited a mild case of COVID-19 infection and presented with symptoms of acute perioral paresthesia and dysarthria. The laboratory analyses indicated a decrease in the concentration of calcium in the blood serum, an elevation in the concentration of phosphorus in the blood serum and a decrease in the concentration of parathyroid hormone (PTH) in the blood serum [73,74,75]. These findings suggest that the infection caused by the SARS-CoV-2 virus led to a significant decrease in calcium levels, particularly in the presence of underlying asymptomatic hypoparathyroidism after surgery [17].

### 4.5. Skeletal Complications and Vertebral Fractures

Morphometric vertebral fractures (VFs) are considered significant clinical manifestations of osteoporosis and skeletal fragility. Recent reports have indicated a high prevalence of these fractures in patients with COVID-19 [76,77,78]. VFs are associated with reduced survival rates, decreased respiratory function and a compromised quality of life in the general population [76].

Hospitalized patients with COVID-19 may experience an elevated risk of fractures due to various concurrent factors [79,80]. These factors involve advanced age and comorbidities like diabetes, cardiovascular diseases and hypertension.

Also, it has been previously reported that individuals who are hospitalized due to COVID-19 regularly show hypovitaminosis D, a condition that is known to be linked to decreased bone mineral density (BMD) and an elevated risk of fractures [45,81].

Vertebral fractures (VFs) and reduced bone mineral density (BMD) have been identified as factors that elevate the likelihood of developing pneumonia and hinder respiratory function, resulting in restrictive pulmonary dysfunction within the general population [82]. VFs have been observed to have an impact on the respiratory function of individuals who have survived COVID-19 in the medium term. This influence, in turn, can have a substantial effect on their overall recovery process and may contribute to the occurrence of long COVID.

### 4.6. Hypomagnesemia

Magnesium deficiency is another important factor that merits attention in the context of COVID-19 and its impact on phosphocalcium metabolism [42]. Magnesium is a vital cofactor in numerous enzymatic reactions, including those involved in calcium regulation. A deficiency in magnesium can disrupt the balance between calcium and phosphorus levels, potentially contributing to the risk of hypocalcemia in post-COVID-19 patients [83].

Several studies have suggested a possible association between magnesium deficiency and COVID-19 severity [83,84,85,86]. COVID-19 patients with severe symptoms often experience significant inflammation and oxidative stress, leading to increased magnesium loss through urine and sweat. Additionally, the use of certain medications during the treatment of COVID-19, such as diuretics and proton pump inhibitors, can further exacerbate magnesium depletion.

Furthermore, magnesium deficiency can interfere with parathyroid hormone (PTH) secretion and action. PTH plays a central role in calcium homeostasis, and impaired PTH function due to magnesium deficiency can result in reduced calcium absorption from the intestine and increased calcium loss through the kidneys [87]. As a consequence, hypocalcemia may ensue, leading to muscle cramps, tingling sensations, confusion and potential cardiac abnormalities.

Considering the interplay between magnesium, calcium and phosphorus, post-COVID-19 patients with magnesium deficiency may be at an increased risk of developing hypocalcemia and associated complications.

## 5. Treatment Studies

References [68,69,70,71] are studies that have investigated the use of vitamin D treatment in patients with COVID-19. As shown, these studies have produced inconsistent results regarding the effectiveness of vitamin D supplementation in improving clinical outcomes or reducing mortality rates in COVID-19 patients.

Entrenas CME, et al. reported improved clinical outcomes and reduced mortality rates with high-dose calcifediol treatment in hospitalized COVID-19 patients. On the other hand, Shah K et al. did not observe a significant improvement in clinical outcomes or mortality rates with high-dose vitamin D treatment in their study.

Rastogi A et al. found limited evidence to support the use of high-dose vitamin D treatment in improving outcomes in hospitalized COVID-19 patients. Similarly, De Niet et al. did not find a significant benefit from vitamin D supplementation in reducing severity or mortality in COVID-19 patients.

These discrepancies in findings highlight the complexity of using vitamin D as a treatment option for COVID-19. The effectiveness of vitamin D supplementation may vary depending on factors such as the stage of the disease, the severity of illness, patient characteristics and dosing regimens. More research is needed to establish clear guidelines on the use of vitamin D in COVID-19 management and to identify patient subgroups that may benefit from this intervention. It is essential for healthcare providers to carefully evaluate the available evidence and consider individual patient factors when making treatment decisions regarding vitamin D supplementation in COVID-19.

## 6. Discussion

The COVID-19 pandemic has not only impacted the respiratory system, but it has demonstrated systemic implications such as modifications in phosphocalcium metabolism. The objective of this review article was to provide a comprehensive summary and critical analysis of the existing research about disorders in phosphocalcium metabolism following COVID-19 [2,26,51]. The review aimed to offer valuable insights into the potential underlying mechanisms and clinical implications associated with these disorders.

The results derived from the examined studies highlight the frequency of phosphocalcium metabolism disorders among individuals in the process of recuperating from COVID-19. Various research studies have documented a variety of disruptions, such as hypophosphatemia, hypocalcemia and secondary hypoparathyroidism [28,50,73]. The results of this study suggest that COVID-19 has the potential to disturb the intricate equilibrium of phosphorus, calcium and parathyroid hormone concentrations within the human body.

Multiple factors contribute to the occurrence of phosphocalcium abnormalities, encompassing the length and intensity of hospitalization, coexisting medical conditions and the occurrence of additional complications like acute respiratory distress syndrome and acute kidney injury (refer to Table 1). There is evidence to suggest that individuals diagnosed with chronic kidney disease or chronic heart disease may be more susceptible to the development of phosphocalcium disorders in the context of COVID-19.

The duration of hospital stay and the severity of the disease are important variables that have a significant impact on phosphocalcium abnormalities in individuals diagnosed with COVID-19. Extended durations of hospitalization, particularly in cases of critical illness, have the potential to disrupt the balance of minerals in the body. This can be attributed to various factors such as reduced ability to move, changes in dietary consumption and the presence of systemic inflammation. Consequently, it is imperative to closely monitor phosphocalcium levels during the entire duration of hospitalization in order to promptly intervene and mitigate complications associated with imbalances in these vital minerals.

One possible mechanism that may contribute to the development of phosphocalcium metabolism disorders following COVID-19 is the direct impact of the virus on the organs responsible for maintaining phosphocalcium balance. Multiple research studies have provided evidence regarding the existence of viral particles and the manifestation of viral receptors within the renal and parathyroid tissues [73,74]. This observation implies that SARS-CoV-2 has the potential to directly impact these organs, resulting in changes to phosphocalcium metabolism.

The COVID-19 infection has the potential to induce systemic inflammation and immune dysregulation, which in turn can have an impact on the regulation of phosphocalcium metabolism. The imbalances of phosphorus, calcium and parathyroid hormone levels can occur because of the activation of the immune system and the release of inflammatory cytokines [2,3]. In addition, the administration of medications in the context of COVID-19 therapy, such as corticosteroids, has the potential to induce metabolic disruptions.

The clinical ramifications of disorders related to phosphocalcium metabolism following a COVID-19 infection are of considerable importance. The presence of hypophosphatemia and hypocalcemia may result in various clinical manifestations, such as muscle weakness, fatigue, bone pain, muscle cramps, tingling sensations and potential cardiac arrhythmias [3,4,26,50].

The effective management of phosphocalcium metabolism disorders in individuals undergoing recovery from COVID-19 necessitates diligent monitoring and the implementation of suitable interventions. Systematic monitoring of phosphorus, calcium and parathyroid hormone concentrations can facilitate the detection and management of any deviations from the optimal levels. The restoration and maintenance of optimal levels may require the utilization of nutritional supplementation, such as vitamin D and calcium. Furthermore, the restoration of phosphocalcium homeostasis could potentially be facilitated by the management of systemic inflammation and immune dysregulation.

Recognizing the constraints of the examined studies holds significance. Numerous studies exhibit limited sample sizes and heterogeneity concerning patient characteristics and methodologies. Additional investigation utilizing larger groups, standardized methodologies and extended monitoring periods is imperative to enhance our comprehension of the frequency, underlying mechanisms and medical consequences associated with disturbances in phosphocalcium metabolism post-COVID-19.

## 7. Conclusions

In conclusion, this review has explained the profound influence of COVID-19 on phosphocalcium metabolism, resulting in a range of metabolic dysfunctions. The results obtained from the examined studies highlight the high occurrence of disruptions in phosphocalcium metabolism among individuals in the process of recovering from COVID-19. These disruptions include hypophosphatemia, hypocalcemia and secondary hypoparathyroidism.

The perturbations in phosphocalcium metabolism following COVID-19 are presumably influenced by a variety of factors, encompassing the direct impact of the virus on the kidneys and parathyroid glands, systemic inflammation, immune dysregulation and the administration of medications during COVID-19 treatment. These mechanisms have the potential to disturb the intricate equilibrium of phosphorus, calcium and parathyroid hormone, leading to imbalances and subsequent clinical manifestations.

The management of disorders related to phosphocalcium metabolism following a COVID-19 infection is of major significance to enhance patient outcomes. The regular monitoring of phosphorus, calcium and parathyroid hormone levels is imperative for the timely identification and implementation of appropriate measures. The restoration and maintenance of optimal levels may require the utilization of nutritional supplementation, such as vitamin D and calcium. The restoration of phosphocalcium homeostasis can be facilitated by addressing systemic inflammation and immune dysregulation.

Nevertheless, it is crucial to recognize the constraints inherent in the examined studies, including limited sample sizes, variability in participant characteristics, and the necessity for additional investigations employing standardized methodologies and extended periods of observation. Future research endeavors should strive to enhance our comprehension regarding the frequency, underlying mechanisms and medical ramifications of phosphocalcium metabolism disorders after the COVID-19 infection.

In summary, this manuscript review highlights the importance of phosphocalcium metabolism disorders in the population recovering from COVID-19. The results underscore the significance of surveillance, interventions and additional investigation in this domain. Healthcare professionals have the potential to enhance the comprehensive care and outcomes of individuals after COVID by addressing these metabolic disturbances.

## 8. Future Perspectives

The review on phosphocalcium metabolism disorders post-COVID-19 provides valuable insights into the impact of SARS-CoV-2 infection on calcium, phosphorus and vitamin D homeostasis in patients. Building on the existing knowledge, several future perspectives can guide further research and clinical practice in this area:

### 8.1. Long-Term Monitoring

Conducting longitudinal studies with extended follow-up periods is essential to understand the long-term consequences of phosphocalcium disorders post-COVID-19. Tracking patients beyond the acute phase will help to assess the persistence of abnormalities and potential late-onset complications, such as osteoporosis and cardiovascular events.

### 8.2. Impact on Bone Health

Investigating the effects of phosphocalcium disorders on bone health is crucial. Long-term studies assessing bone mineral density, bone turnover markers and fracture risk in post-COVID-19 patients can provide valuable insights into bone-related complications and guide preventive measures.

### 8.3. Optimal Vitamin D Supplementation

Conducting randomized controlled trials to determine the most effective and safe dosage of vitamin D supplementation in post-COVID-19 patients is essential. Understanding the optimal timing, duration and formulation of supplementation can improve patient outcomes and reduce complications.

### 8.4. Immune System Dysregulation

Exploring the immunological mechanisms underlying phosphocalcium metabolism disorders post-COVID-19 is critical. Investigating the role of immune dysregulation, cytokine storm and chronic inflammation in disrupting calcium and phosphorus homeostasis can provide potential therapeutic targets.

### 8.5. Multidisciplinary Care Teams

Establishing multidisciplinary care teams involving endocrinologists, nephrologists, infectious disease specialists and rehabilitation experts can provide comprehensive management for post-COVID-19 patients with phosphocalcium disorders. Collaboration among specialties can address the complexity of these conditions.

### 8.6. Global Collaboration and Data Sharing

Encouraging international collaboration and data sharing among researchers and institutions can enhance the collective understanding of phosphocalcium disorders post-COVID-19. Large-scale multinational studies can yield robust findings and facilitate more comprehensive guidelines.

### 8.7. Preparing for Future Outbreaks

Applying the knowledge gained from studying phosphocalcium disorders post-COVID-19 can help healthcare systems to prepare for future infectious disease outbreaks. Lessons learned from managing these disorders can inform the strategies for preventing and managing similar complications in future pandemics.

In conclusion, the future perspectives outlined above offer a roadmap for furthering our understanding of phosphocalcium metabolism disorders in the context of COVID-19 recovery. By addressing these perspectives through dedicated research and collaboration, we can optimize patient care, mitigate long-term consequences and enhance public health measures to improve the overall well-being of post-COVID-19 patients.

## Figures and Tables

**Figure 1 biomedicines-11-02362-f001:**
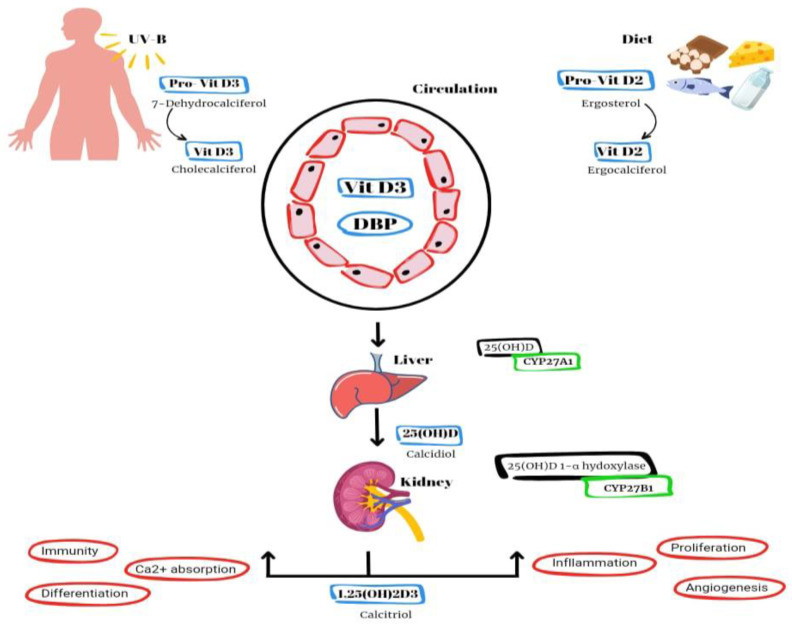
Schematic illustration of the vitamin D synthesis pathway and signaling mechanism. The primary variants of vitamin D found in the natural environment encompass vitamin D3 (cholecalciferol), which is biosynthesized in the epidermis of animals and humans upon exposure to sunlight, as well as acquired through dietary source (DBP = vitamin D-binding protein).

**Figure 2 biomedicines-11-02362-f002:**
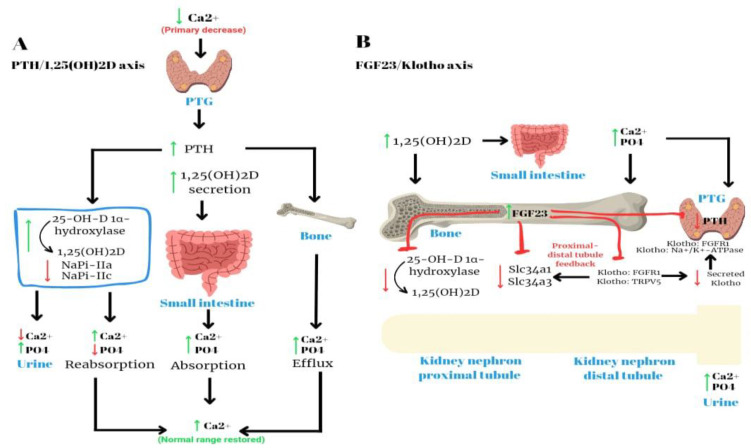
The maintenance of calcium–phosphate balance is controlled by the parathyroid glands, bone, kidneys and intestinal tract.

**Figure 3 biomedicines-11-02362-f003:**
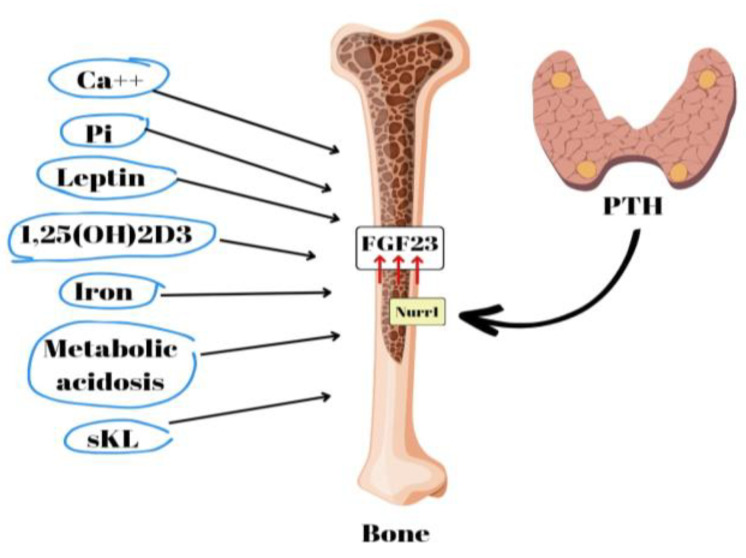
A schematic illustration of several factors that have been identified as FGF23 production inducers. Calcium (Ca^++^), phosphate (Pi), leptin, secreted Klotho (sKL), iron, and metabolic acidosis.

**Figure 4 biomedicines-11-02362-f004:**
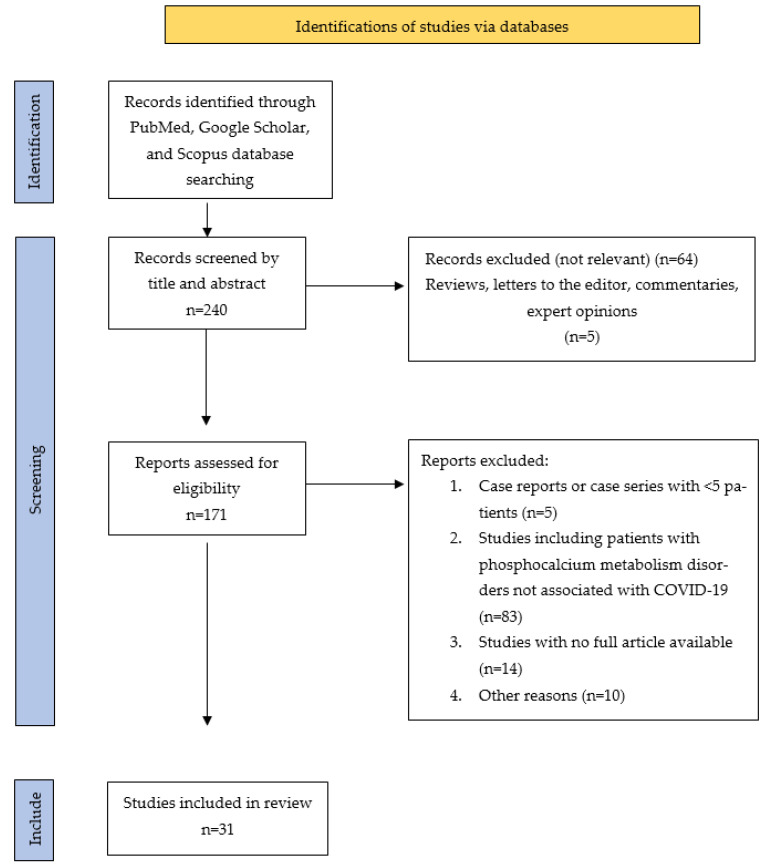
Flow diagram of published studies related to COVID-19-associated phosphocalcium metabolism disorders.

**Figure 5 biomedicines-11-02362-f005:**
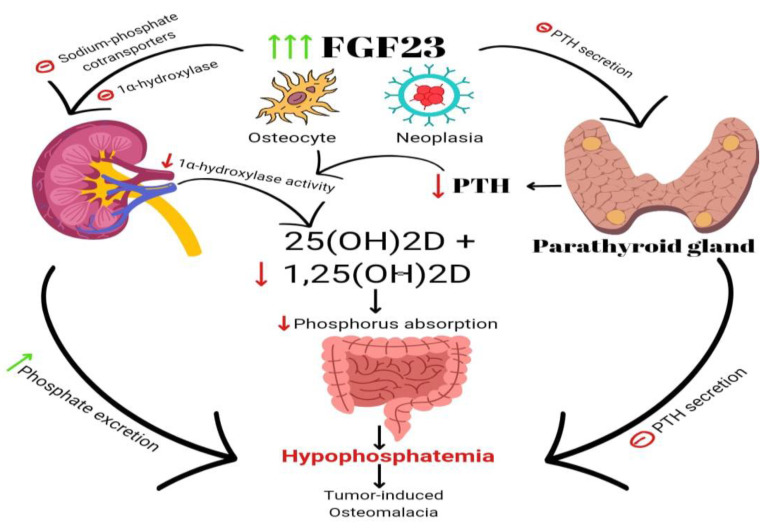
Relationship between serum phosphate levels and vitamin D, PTH and calcium.

**Table 1 biomedicines-11-02362-t001:** Phosphocalcium metabolism abnormalities and post-COVID-19 complications: a comprehensive review of studies.

Study	Study Design	Patients (n)	Age (Mean ± SD)	COVID-19 Severity	Duration of Hospitalization (Days)	Phosphocalcium Disorders	Other Complications	Comorbidity	Country
Yiquin W et al. [29]	Retrospective	125	55 ± 8.9 years	Mild	13 ± 3.9	Hypocalcemia: 64%	Pneumonia	Diabetes type 2 Hypertension Coronary heart disease	China
Bálint D et al. [30]	Multicenter study	451	58.2 ± 10.3 years	Severe	14 ± 3.9	Hypocalcemia: 45%	Severe respiratory failures	Cirrhosis	Hungary
Wessam O et al. [31]	Prospective	445	50 ± 12.1 years	Mild	7.5 ± 2.1	Hypocalcemia: 75% Hypovitaminosis D: 5%	Chronic respiratory diseases	Diabetes Hypertension	Oman
Meera M et al. [21]	Retrospective	506	65 ± 7.2 years	Severe	25.6 ± 4.7	Hypocalcemia: 53%	Lymphopenia Hypoxia	Obesity Hypertension Cancer	England
Berta T et al. [32]	Retrospective	316	65 ± 9.5 years	Severe	19.3 ± 3.5	Hypocalcemia: 63%	ICU admission	Cardiopathy Dyslipidemia Diabetes	Spain
Jyot A et al. [33]	Prospective	170	62 ± 11.7 years	Critical	23.7 ± 5.2	Hypocalcemia: 80%	Acute respiratory distress syndrome	Hypertension Diabetes mellitus type 2	India
Jingmei L et al. [19]	Prospective	69	68 ± 6.8 years	Severe	14.1 ± 1.9	Hypocalcemia: 62%	Pneumonia	Coronary heart disease Hypertension	China
Rourang W et al. [4]	Case series	435	57 ± 10.5 years	Severe	10 ± 4	Hypophosphatemia: 7.6%	Chronic liver disease	COPD Hypertension Diabetes mellitus	China
Zijin C et al. [34]	Retrospective	823	60.9 ± 14.9 years	Severe	14.8 ± 6.3	Hypophosphatemia: 10%	Acute liver injury	COPD Diabetes mellitus	Hungary
Hannah W et al. [35]	Cross-sectional	1226	61.2 ± 8.3 years	Severe	13.5 ± 2.8	Hypophosphatemia: 26%	Pneumonia	Cardiovascular disease Septic shock	Italy
Marina V et al. [36]	Prospective	104	59 ± 14 years	Critical	20.3 ± 9	Hypophosphatemia: 33%	Gastrointestinal problems	Diabetes Obesity	Switzerland
Hadavi M et al. [37]	Retrospective	1346	65.9 ± 1.1 years	Severe	14 ± 6	Hypophosphatemia: 10%	Respiratory failure	Cardiovascular disease Hypertension Autoimmune disease	Iran
Kormann R et al. [38]	Retrospective	42	63.4 ± 8 years	Severe	19 ± 12.2	Hypophosphatemia: 29%	Kidney disease	Hypertension Dyslipidemia COPD	France
Charles L et al. [39]	Retrospective	83	58 ± 12.7 years	Severe	14 ± 3.1	Hypophosphatemia: 18%	Gastrointestinal problems	Hypertension Diabetes Cancer	Singapore

Note: The table presents a comprehensive summary of studies investigating post-COVID-19 phosphocalcium metabolism changes. The studies are categorized based on their study design, and each row provides essential information, including the number of patients, age range, COVID-19 severity, duration of hospitalization, phosphocalcium findings, other complications, comorbidities and the country where the study was conducted. The table serves as a valuable reference to understand the impact of COVID-19 on phosphocalcium metabolism and associated complications.

**Table 2 biomedicines-11-02362-t002:** Prevalence of hypovitaminosis D and phosphocalcium abnormalities in post-COVID-19 patients.

Study	Study Design	Patients (n)	Age (Mean ± SD)	COVID-19 Severity	Phosphocalcium Findings	Country	Affected Organs/Systems
Saponaro F et al. [58]	Case series	93	68 ± 16 years	Mild	Hypovitaminosis D: 65%	Italy	Lungs
Luigi di F et al. [5]	Case series	78	48.5–64.5 years	Severe	Hypovitaminosis D: 67.9% Hypocalcemia: 70.5%	Italy	Parathyroid
Singh S et al. [59]	Retrospective	360	35.63 years	Mild	Hypovitaminosis D: 16%	India	Lungs
Mohammed A et al. [60]	Cross-sectional study	124	50 years	Severe	Hypovitaminosis D: 97.6%	Egypt	Lungs
Manojlovic M et al. [61]	Retrospective	74	57.64 ± 17.83 years	Severe	Hypovitaminosis D: 44.6% Vitamin D insufficiency: 8.1%	Serbia	Heart
Mazziotti G et al. [62]	Case series	348	26.0–95.0 years	Mild	Hypovitaminosis D: 46.3%	Italy	Parathyroid
D’Alessandro A et al. [63]	Multicenter study	163	65 ± 13 years	Severe	Hypovitaminosis D: 82%	Italy	Lungs
Rimesh P et al. [26]	Retrospective case	72	37.5 ± 13.7 years	Mild	Hypovitaminosis D: 97% Hypocalcemia: 67%	India	Bones
Rizaldy P et al. [64]	Case series	10	49.6 years	Mild	Hypovitaminosis D: 90%	Indonesia	Heart
Carpagnano G et al. [65]	Retrospective	42	65 ± 13 years	Severe	Hypovitaminosis D: 81%	Italy	Lungs
Thiago J et al. [66]	Cross-sectional	176	72.9 ± 9.1 years	Severe	Hypovitaminosis D: 93.8%	Brazil	Heart
Nascimento R et al. [67]	Cross-sectional	1478	35.2 years	Mild	Hypovitaminosis D: 18.2%	Brazil	Heart
Entrenas C et al. [68]	Clinical trial	76	53 ± 10 years	Mild	Hypovitaminosis D	Spain	Lungs
Shah K et al. [69]	Meta-analysis	532	N/A	Mild	Hypovitaminosis D	India	Lungs
Rastogi et al. [70]	Randomized, placebo-controlled	40	50 years	Mild	Hypovitaminosis D	India	Heart
De Niet et al. [71]	Placebo-controlled trial	50	63.24 ± 14.46 years	Mild	Hypovitaminosis D	Belgium	Kidney

## Data Availability

Data are contained within the article.

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
