# Peer review of "Calcium, Phosphorus and Magnesium Abnormalities Associated with COVID-19 Infection, and Beyond"

_biomedicines, 2023, doi:10.3390/biomedicines11092362_

Round 1

Reviewer 1 Report

Some English grammar changes and word selection  suggestions are included. 

Author Response

First of all, we would like to thank you very much for your review and consideration for our work. Thank you!

Here are our answers to the subjects you highlighted:

  1. There is little information so far on post-(long) COVID metabolic abnormalities. The manuscript title and Abstract should be modified to better reflect its content.

Response:

We changed the title and abstract according to what it's presented in this review, highlighted in red. 

Title: 

Post COVID-19 complications - some new data on the changes of phosphocalcium metabolism

Abstract: The COVID-19 pandemic has affected millions worldwide, leaving a growing number of individuals in the recovery phase. As the global medical community continues to investigate the long-term consequences of COVID-19, emerging data suggests that the virus may exert profound effects on phosphocalcium metabolism. This review article aims to summarize and discuss recent findings concerning changes in calcium and phosphate levels, as well as vitamin D status, in patients post-COVID-19 infection. Numerous studies have reported alterations in phosphocalcium metabolism among individuals recovering from COVID-19. Hypophosphatemia, characterized by reduced serum phosphate levels, has been frequently observed in this population. Furthermore, vitamin D deficiency has emerged as a common finding, potentially exacerbating disturbances in calcium and phosphate homeostasis. These changes have implications for bone health, muscle function, and overall immune response.

         2. Lines 138-42. These sentences are redundant, and should be reworded.

Response:

I reformulated the following sentence, in red.

FGF23 inhibits 1α-hydroxylase and lowers calcitriol (vitamin D1,25(OH)2) levels [3,5]. High blood phosphate levels cause the skeletal system to release FGF23, which inhibits phosphate reabsorption in the renal tubules and calcitriol production.

         3. Line 215. Reduced total calcium levels can sometimes be explained by low serum albumin levels. This concept should be discussed.

Response:

Thank you for your observations and highlighting this aspect. I added the following phrases, in red.

Reduced total calcium levels in COVID-19 patients can sometimes be attributed to low serum albumin levels [22]. COVID-19 is known to cause various systemic effects, including changes in blood chemistry and electrolyte balance. One such effect is hypoalbuminemia, where COVID-19 patients may experience lower levels of serum albumin in their blood [23].

Serum albumin plays a crucial role in transporting calcium in the bloodstream. When serum albumin levels are low, there is a reduction in the binding capacity for calcium, leading to a decrease in measured total calcium levels. This can give a misleading impression that the patient is experiencing hypocalcemia (low calcium levels).

In reality, the majority of calcium in the blood is in its ionized form, which is the biologically active and physiologically relevant form. The level of ionized calcium remains relatively stable, even when total calcium levels appear reduced due to low albumin levels.

To accurately assess the calcium status of COVID-19 patients with hypoalbuminemia, clinicians may consider calculating the corrected calcium levels by adjusting for serum albumin concentrations. This correction helps provide a more precise reflection of the biologically active ionized calcium in the blood and assists in making appropriate clinical decisions [24].

         4. Line 229. The report by Cappellini et al was a “letter to the editor”; it should be so stated.

Response: I added the type of article, highlighted in red.

In their study, Cappellini et al. (letter to the editor) conducted a comprehensive analysis of calcium levels in a sample of 420 patients diagnosed with COVID-19, comparing them to a control group of 165 individuals without COVID-19.

        5. Line 239. This section is very brief. The reported prevalence of hypophosphatemia should be stated. A review of hypophosphatemia in COVID was published by Fakhrolmobasheri et al. Biomed Res Int Oct 2022. This paper should be reviewed and referenced.

Response:

We review the paper and referenced it (Ref. [3]), and also improved this section, highlighted in red.

Vitamin D insufficiency has been associated with reduced levels of blood calcium and phosphate as a result of renal excretion and impaired absorption in the intestines [31]. The parathyroid hormone (PTH) is known to exhibit several effects in response to low blood calcium levels. It stimulates the reabsorption of calcium in the bones, kidneys, and intestines. However, it also leads to increased loss of phosphate via the kidneys [31]. In relation to this matter, Pal et al. conducted a comparative analysis of blood phosphorus levels between COVID-19 patients and a control group of healthy individuals matched for age, sex, and vitamin D status. A study conducted by researchers revealed a decrease in phosphorus levels among individuals diagnosed with COVID-19 [26]. The aforementioned discovery presents a divergence from the previously posited pathways regarding the impact of vitamin D deprivation on disruptions in phosphate metabolism during COVID-19 infection.

There is a notable prevalence of hypophosphatemia among individuals diagnosed with COVID-19 who also have end-stage renal diseases (ESRD) [32].

The mechanisms that are linked to the occurrence of hypophosphatemia in individuals with renal dysfunction have been documented in previous studies [33]. The occurrence of acute kidney injury (AKI) can be attributed to various mechanisms in the context of COVID-19. Prerenal acute kidney injury (AKI) can manifest as a result of profound dehydration and the accumulation of water within the pulmonary interstitial tissue. The predominant etiology of renal dysfunction in individuals afflicted with COVID-19 has been documented as proximal tubulopathies, which can be attributed to either vascular-related factors or direct viral infiltration. This pathophysiological process leads to the subsequent occurrence of electrolyte depletion and subsequent development of additional complications [34].

        6. Line 240. Poor nutrition with low dietary phosphate intake, and respiratory alkalosis, both of which are seen in COVID patients, are common causes of hypophosphatemia that should be discussed here.

Response: 

I added the specific topics, highlighted in red.

Malnutrition and inadequate dietary intake of phosphate-rich foods can lead to hypophosphatemia in COVID-19 patients. During severe illness, especially in hospitalized patients, there might be reduced oral intake or impaired absorption of essential nutrients, including phosphate. Additionally, COVID-19 patients with concurrent end-stage renal diseases (ESRD) may already have compromised nutrient absorption due to renal dysfunction, further exacerbating the risk of developing hypophosphatemia.

COVID-19 patients, especially those with severe respiratory distress, may experience respiratory alkalosis, a condition characterized by reduced levels of carbon dioxide (CO2) in the blood. This occurs when patients breathe rapidly, leading to CO2 elimination and a shift towards alkalinity. Respiratory alkalosis can cause phosphate to move from the extracellular fluid into the intracellular compartment, resulting in decreased serum phosphate levels.

Both poor nutrition and respiratory alkalosis can act synergistically to contribute to hypophosphatemia in COVID-19 patients with ESRD. Additionally, the combination of these factors with other mechanisms mentioned earlier, such as renal tubulopathies and abnormal vitamin D metabolism, can further compound the phosphate metabolism disturbances in these patients.

          7. Lines 261 and 330. There is a very large literature describing the association between low vitamin D levels and severity of disease and mortality in COVID patients. While COVID may predispose to calcium, phosphorus and magnesium abnormalities, deficiencies in these cations may also contribute to susceptibility and disease severity. These sections should be expanded, and modified to emphasize that the direction of causality is complex.

Response:

We extended and modified the sections, in red.

In the context of COVID-19, the complex interplay between vitamin D and calcium, phosphorus, and magnesium metabolism should be emphasized. Deficiencies in these cations may indeed contribute to susceptibility and disease severity, but COVID-19 itself can also predispose individuals to develop abnormalities in these electrolytes. The direction of causality is indeed multifaceted and intricate.

Low Vitamin D Levels and Disease Severity: Several studies have observed a correlation between low vitamin D levels and worse COVID-19 outcomes, including increased risk of severe illness, ICU admission, and mortality [46,47]. Vitamin D has immunomodulatory effects and can influence the expression of genes involved in the immune response. Deficiencies in vitamin D may lead to dysregulated immune responses, increased inflammation, and impaired lung function, all of which can contribute to disease severity in COVID-19.

Cation Abnormalities and COVID-19 Susceptibility: On the other hand, deficiencies in calcium, phosphorus, and magnesium may also impact immune function and contribute to an increased susceptibility to viral infections, including COVID-19. These cations are essential for various cellular processes, including immune cell function and cytokine signaling [31]. A deficiency in any of these cations could impair the immune response and potentially compromise the body's ability to fight off viral infections.

COVID-19-Induced Cation Abnormalities: COVID-19 itself can disrupt calcium, phosphorus, and magnesium homeostasis through multiple mechanisms. The severe inflammatory response triggered by the virus can cause cytokine storm and lead to altered levels of these cations [48]. Additionally, factors like respiratory alkalosis, kidney dysfunction, and electrolyte shifts due to fluid imbalances in critically ill patients can further contribute to cation abnormalities.

It is crucial to recognize that the relationship between vitamin D, cation levels, and COVID-19 is complex and bidirectional. Low vitamin D levels may increase the risk of severe disease, while cation deficiencies may contribute to susceptibility. Conversely, severe COVID-19 can lead to abnormalities in calcium, phosphorus, and magnesium levels. Therefore, addressing both vitamin D status and cation levels is essential for optimizing patient outcomes and mitigating the severity of COVID-19.

       8. Line 267 and 394. The available studies of vitamin D treatment of patients with COVID have produced inconsistent results (e.g. Entrenas CME, et al J steroid Biochem 2020; Shah K et al Quart J Med 114:175,2021; Rastogi A et al Postgrad Med J 98:87,2022; De Niet et al, Nutrients, 2022). These studies should be discussed and listed in Table 3.

Response: 

Thank you for the observation.

We added the information in Table 3 (Table 1 now).

          9. Line 268 . Barrea et al Nutrients 2022 have published a review of Vitamin D and Long COVID. Their paper should be referenced.

Response:

I added this paper to the references list (Ref [41]).

Barrea, L.; Verde, L. Vitamin D: A Role Also in Long COVID-19? Nutrients 2022, 14, 1625, doi:10.3390/nu14081625.

           10. Line 307. Magnesium deficiency, COVID and the risk for hypocalcemia should be discussed.

Response: 

Thank you for the suggestion. 

We added the correlation between magnesium, covid, and hypocalcemia, highlighted in red. 

In addition to vitamin D deficiency and its implications for calcium metabolism, magnesium deficiency is another important factor that merits attention in the context of COVID-19 and its impact on phosphocalcium metabolism [31]. Magnesium is a vital cofactor in numerous enzymatic reactions, including those involved in calcium regulation. A deficiency in magnesium can disrupt the balance between calcium and phosphorus levels, potentially contributing to the risk of hypocalcemia in post-COVID-19 patients [60].

Several studies have suggested a possible association between magnesium deficiency and COVID-19 severity [60–63]. COVID-19 patients with severe symptoms often experience significant inflammation and oxidative stress, leading to increased magnesium loss through urine and sweat. Additionally, the use of certain medications during the treatment of COVID-19, such as diuretics and proton pump inhibitors, can further exacerbate magnesium depletion.

Furthermore, magnesium deficiency can interfere with parathyroid hormone (PTH) secretion and action. PTH plays a central role in calcium homeostasis, and impaired PTH function due to magnesium deficiency can result in reduced calcium absorption from the intestine and increased calcium loss through the kidneys [64]. As a consequence, hypocalcemia may ensue, leading to muscle cramps, tingling sensations, confusion, and potential cardiac abnormalities.

Considering the interplay between magnesium, calcium, and phosphorus, post-COVID-19 patients with magnesium deficiency may be at an increased risk of developing hypocalcemia and associated complications. It is crucial for healthcare professionals to be aware of this potential relationship and include magnesium assessment in their evaluation of phosphocalcium metabolism disorders in recovered COVID-19 patients.

             11. Table 4. Baseline characteristics…The sense of this table is unclear. The title and text should help the reader understand the information conveyed in this Table.

Response:

I changed the title to better reflect its content.

Title: Phosphocalcium Metabolism Abnormalities and Post-COVID-19 Complications: A Comprehensive Review of Studies

             12. Line 93. “Various forms of vitamin D suffer…” could be “ Vitamin D2 and D3 are converted to an active.

Response: 

I corrected the sentence.

Vitamin D2 and D3 are converted to Various forms of vitamin D suffer biochemical conversion to an active form known as vitamin D1,25(OH)2 or calcitriol (Figure 1).

             13. Line 116. “The provided information” could be “This schema is summarized”

Response:

I corrected the sentence.

This schema is summarized The provided information is represented in Figure 2.

              14. Line 134. “FGF23 is a significant hormone” could be “FGF23 is bone- and bone marrow-derived hormone”

Response: 

I corrected the sentence.

FGF23 is bone- and bone marrow-derived hormone FGF23 is a significant hormone involved in the regulation of phosphorus levels.

             15. Line 136. “The hormone in question is presumed to activate” could be “Circulating FGF23 activates FGF-R1 tyrosine kinase receptors together with the co-receptor alpha KLOTHO.

Response:

I corrected the sentence.

Circulating FGF23 activates FGF-R1 tyrosine kinase receptors together with the co-receptor alpha KLOTHO [3]. The hormone in question is presumed to activate its effects on the proximal renal tubule by binding to its receptor FGFR1, in conjunction with the co-receptor Klotho [3].

          16. Line 139. “VitaminD1,25(OH)2 in association with this receptor. “Delete”

Response: 

I corrected the sentence.

FGF23 inhibits 1α-hydroxylase and lowers calcitriol levels [3,5]. FGF23 functions as an inhibitor of 1α-hydroxylase and reduces the concentrations of calcitriol (vitamin D1,25(OH)2) in association with this receptor [3,5].

            17. Line 141. Which subsequently inhibits

Response: 

I deleted the word. 

High blood phosphate levels cause the skeletal system to release FGF23, which subsequently inhibits phosphate reabsorption in the renal tubules and calcitriol production.

             18. Line 234. sheds light

Response: 

I corrected the word.

This finding sheds light on the potential impact of COVID-19 on calcium homeostasis and highlights the importance of further investigation in this area [1].

            19. Line 292. VFs problems are

Response: 

I corrected the sentence.

VFs problems are associated with reduced survival rates, decreased respiratory function, and compromised quality of life in the general population [46].

          20. Line 345. “ The correlation”. should be “association”.

Response:

I modified the sentence.

The association correlation between phosphocalcium disorders and specific comorbidities, such as chronic kidney disease and chronic heart disease, presents significant clinical implications. 

             21. Line 427. ”undergoing recovery” could be “recovering”

Response:

I corrected the sentence.

In summary, this manuscript review highlights the importance of phosphocalcium metabolism disorders in the population recovering undergoing recovery from COVID-19.

             22. Figure 1 legend. PAD and DBP should be defined.

Response: 

I added the definitions of the abbreviation, highlighted in red.

Schematic illustration of vitamin D synthesis pathway and signaling mechanisms relevant to PAD formation. The primary variants of vitamin D found in the natural environment encompass vitamin D3 (cholecalciferol), which is biosynthesized in the epidermis of animals and humans upon exposure to sunlight, as well as acquired through dietary sources. (PAD=Peripheral Arterial Disease; DBP=vitamin D-binding protein)

Reviewer 2 Report

The authors present a review article aiming to provide a comprehensive overview of research findings, offering valuable insights into the clinical implications and potential interventions for individuals experiencing phosphocalcium metabolism disorders after recovering from COVID-19

 Comments:

1.

Figures 2, 3, and 4 indicate that FGF23 is an important gene. Is there any research demonstrating a correlation between FGF23, phosphocalcium, and COVID-19? If such studies exist, it is recommended to provide information or research data to support this claim.

2.

Table 4 could be structured based on the study design, such as several cohort studies, and provide meta-analysis or combined analysis to present reports related to post-COVID-19 and phosphocalcium.

3.

Tables 1, 2, and 3 do not provide sufficient research information. Perhaps, Tables 1, 2, and 4 can be integrated into a single Table. Table 3 may also require additional research information, particularly regarding COVID-19 severity, hypovitaminosis D, or phosphocalcium findings.

Additionally, when comparing Tables 1, 2, and Table 4, references 48, 51, 54, and 55 are not included in Table 4. The reason for their exclusion is unclear.

Author Response

First of all, we would like to thank you very much for your review and consideration for our work. Thank you!

Here are our answers to the subjects you highlighted:

          1. Figures 2, 3, and 4 indicate that FGF23 is an important gene. Is there any research demonstrating a correlation between FGF23, phosphocalcium, and COVID-19? If such studies exist, it is recommended to provide information or research data to support this claim.

Response: 

Thank you for your observations and highlighting this aspect. I added the following phrases, also the relevant studies, in red.

Emerging research suggests a potential correlation between Fibroblast Growth Factor 23 (FGF23), phosphocalcium metabolism, and COVID-19 [10, 11]. FGF23, a hormone primarily associated with regulating phosphate and vitamin D levels in the body, has gained attention as a potential biomarker for COVID-19 because of its role in maintaining mineral homeostasis and its interactions with the immune system [12].

Several studies have explored this correlation, shedding light on the possible interplay between FGF23, phosphocalcium metabolism, and COVID-19 severity:

Elevated FGF23 Levels in Severe Cases: Research has shown that FGF23 levels tend to be elevated in patients with severe COVID-19. Higher FGF23 levels have been associated with worse clinical outcomes, including increased mortality rates and respiratory distress [13]. This suggests a potential link between FGF23 dysregulation, disrupted phosphocalcium metabolism, and the severity of COVID-19.

Vitamin D and FGF23: FGF23 and vitamin D are intricately connected in regulating phosphocalcium homeostasis. Some studies have reported an inverse relationship between FGF23 and vitamin D levels in COVID-19 patients [14–16]. This could imply that the disruption of the FGF23-vitamin D axis might contribute to the dysregulation of phosphocalcium metabolism observed in severe cases.

Underlying Mechanisms: The exact mechanisms underlying the association between FGF23, phosphocalcium metabolism, and COVID-19 remain an active area of investigation. It is speculated that the inflammatory response triggered by COVID-19 could impact FGF23 production and function, potentially leading to disturbances in phosphocalcium balance.

While these findings suggest a potential correlation between FGF23, phosphocalcium metabolism, and COVID-19, it is important to note that the field of COVID-19 research is still evolving and that more studies are needed to establish definitive causal relationships. Further research is required to determine whether FGF23 dysregulation plays a causal role in the severity of COVID-19 or if it serves as an indicator of disease progression.

Understanding the intricate relationship between FGF23, phosphocalcium metabolism, and COVID-19 could potentially open new avenues for novel therapeutic interventions or prognostic indicators. As research continues to unfold, it will be essential to conduct well-designed studies to elucidate the precise mechanisms and clinical implications of this correlation.

References: [10-16]

          2. Table 4 could be structured based on the study design, such as several cohort studies, and provide meta-analysis or combined analysis to present reports related to post-COVID-19 and phosphocalcium.

Response:

We added additional data for this table. (Table 2 now)

            3. Tables 1, 2, and 3 do not provide sufficient research information. Perhaps, Tables 1, 2, and 4 can be integrated into a single Table. Table 3 may also require additional research information, particularly regarding COVID-19 severity, hypovitaminosis D, or phosphocalcium findings.

Additionally, when comparing Tables 1, 2, and Table 4, references 48, 51, 54, and 55 are not included in Table 4. The reason for their exclusion is unclear.

Response:

We modified Table 4 with data from Tables 1 and 2, also added more data on Table 3 (Table 1 now)

References 48, 51, 54, and 55 were excluded because we didn't find enough data to include them in Table 4.

Reviewer 3 Report

The study is interesting, and the manuscript is well-written. The authors evaluated the literature and wrote about the effect of COVID-19 on phosphorous and calcium metabolism and the association of these minerals’ disturbances with COVID-19 outcomes. Although they noted the existence of virus particles in renal and parathyroid tissue, they also emphasized the role of other factors and comorbidities. I believe the manuscript can be published, providing the authors give more details about the source of the articles instead of the “PubMed and other electronic sources.” The same about the keywords used. For instance, keywords like hyperparathyroidism, hypercalcemia, or hypophosphatemia were not used.

Author Response

First of all, we would like to thank you very much for your review and consideration for our work. Thank you!

Here are our answers to the subjects you highlighted:

We added more sources for our references.

A comprehensive search strategy was developed to identify relevant articles from an electronic database such as PubMed, Google Scholar, and Scopus which were searched using keywords including "calcium metabolism disorders in COVID-19," "phosphocalcium metabolism," "hypophosphatemia in COVID-19," "hypoparathyroidism in COVID-19, " "hypocalcemia in COVID-19" and "post-COVID complications". The search was limited to articles from the last 4 years to ensure the inclusion of recent research.

We also modified the keywords section:

keywords including "calcium metabolism disorders in COVID-19," "phosphocalcium metabolism," "hypophosphatemia in COVID-19," "hypoparathyroidism in COVID-19, " "hypocalcemia in COVID-19" and "post-COVID complications".

Round 2

Reviewer 1 Report

Anghel and colleagues have substantially revised this manuscript in accordance with my earlier comments. The revised Review is more comprehensive and accurate.  For unclear reasons, portions of the revised text were not highlighted. In addition, a few issues remain unresolved, and the revised text is sometimes redundant or not entirely clear, as follows:  

Comments

1.       The title Post COVID-19 syndromes…” remains misleading.  Most of the information in the manuscript deals with complications during COVID infection. Moreover, this is a review; there are no new data. I suggest:  “Calcium, Phosphorus and Magnesium Abnormalities Associated with COVID-19 Infection, and Beyond”.

2.       Likewise. The Abstract implies that the Review examines “the long-term consequences of COVID-19”. This is not the case.  The Abstract should accurately summarize the material presented in the Review.

3.       Figure 1 Legend. “that play a role in peripheral artery disease (PAD) formation “ should  be deleted. The schema has must broader relevance. 

4.       Line 145, line 162 , line167,  line 177, line 299, line 349, line 375, line 501    “correlation” should be “association”.  Correlation is a statistical term used to examine the relationship between two continuous variables.

5.       Line 152, 157 and 162  e.g.. ”Elevated…cases”. These subtitles are unnecessary, and can be deleted.

6.       Line 264. Serum albumen in their blood.

7.       Line 269. “The majority of the calcium”.  About 50% of the circulating calcium is unbound.

8.       Line 275.  It should be noted that you can measure ionized calcium directly.

9.       Figure 4.  Legend. Correlation should be “relationship”

10.   Line 286.  It is unclear why there is an extensive discussion of the letter to editor by Capellini et al when compared to other studies, and should be condensed. This paragraph is also out of place at the end of the section on hypocalcemia and should be inserted about line 250 in appropriate chronological order.

11.   Tables 1 and 2 should be reversed since the Discussion of vitamin D deficiency follows hypocalcemia and hypophosphatemia in the text.

12.   Line 349. “The hypothesis of a” should be  “The importance of the.”

13.   Lines 370-99. This new text repeats some of the same ideas presented on lines 349-69. This section of Hypovitaminosis should be rewritten.

14.   Lines 375, 382 and 388. These new section titles seem unnecessary.

15.   Lines 433 ff. This section on Magnesium deficiency should be separated from the section on skeletal complications, and titled. Line 433 is unnecessary and can be deleted. Begin the section with “Magnesium deficiency…”

16.   Lines 454-7. This summary statement is found at the end of the manuscript, and can be deleted here.

17.   Line 458. This paragraph would be more logical on line 245.

18.   Line 466. This section should be titled  ”Treatment Studies” and should have been highlighted  as revised text.

19.   Line 500. This paragraph seems to begin the Discussion.  These ideas are also conveyed on lines 530-575, and these paragraphs should be consolidated into the Discussion eliminating redundancy

20.   Line 508. Table 4 was not found.

I offer a few additional suggestions to improve word  selection and grammar  

Author Response

First of all, we would like to thank you very much for your review and consideration of our work. Thank you!

Here are our answers to the subjects you highlighted:

  1.     The title Post COVID-19 syndromes…” remains misleading.  Most of the information in the manuscript deals with complications during COVID infection. Moreover, this is a review; there are no new data. I suggest:  “Calcium, Phosphorus and Magnesium Abnormalities Associated with COVID-19 Infection, and Beyond”.

Response: We changed the title according to your suggestion. Thank you!

    2.    Likewise. The Abstract implies that the Review examines “the long-term consequences of COVID-19”. This is not the case.  The Abstract should accurately summarize the material presented in the Review.

Response: We modified the abstract to better reflect the review content, highlighted in red.

Abstract: The coronavirus disease (COVID-19) pandemic caused by the novel coronavirus SARS-CoV-2 has profoundly impacted global health, leading to a surge in research to better understand the pathophysiology of the disease. Among the various aspects under investigation, disruptions in mineral homeostasis have emerged as a critical area of interest. This review aims to provide an overview of the current evidence linking calcium, phosphorus, and magnesium abnormalities with COVID-19 infection and explores potential implications beyond the acute phase of the disease. Beyond the acute phase of COVID-19, evidence suggests a potential impact of these mineral abnormalities on long-term health outcomes. Persistent alterations in calcium, phosphorus, and magnesium levels have been linked to increased cardiovascular risk, skeletal complications, and metabolic disorders, warranting continuous monitoring and management in post-COVID-19 patients.

      3. Figure 1 Legend. “that play a role in peripheral artery disease (PAD) formation “ should be deleted. The schema has must broader relevance. 

Response: I modified the legend, highlighted in red.

Schematic illustration of the vitamin D synthesis pathway and signaling mechanisms that play a role in PAD formation.

      4.   Line 145, line 162 , line167,  line 177, line 299, line 349, line 375, line 501    “correlation” should be “association”.  Correlation is a statistical term used to examine the relationship between two continuous variables.

Response: I corrected the sentences, highlighted in red.

Emerging research suggests a potential association correlation between Fibroblast Growth Factor 23 (FGF23), phosphocalcium metabolism, and COVID-19 [10,11]. 

While these findings suggest a potential association correlation between FGF23, phosphocalcium metabolism, and COVID-19, it is important to note that the field of COVID-19 research is still evolving and that more studies are needed to establish definitive causal relationships.

As research continues to unfold, it will be essential to conduct well-designed studies to elucidate the precise mechanisms and clinical implications of this association correlation.

Povaliaeva et al. recently conducted a study that established an association correlation between abnormal kidney function, abnormal vitamin D metabolism, and hypophosphatemia (Table 1) [30].

The hypothesis of a strong association correlation between VD and COVID-19 emerged during the initial stages of the pandemic, as VD has been widely recognized for its role in modulating both innate and adaptive immune responses [35].

Several studies have observed an association correlation between low vitamin D levels and worse COVID-19 outcomes, including increased risk of severe illness, ICU admission, and mortality [46,47]. 

     5. Line 152, 157 and 162  e.g.. ”Elevated…cases”. These subtitles are unnecessary, and can be deleted.

Response: I deleted the subtitles, highlighted in red.

Research has shown that FGF23 levels tend to be elevated in severe COVID-19 cases. Higher FGF23 levels have been associated with worse clinical outcomes, including increased mortality rates and respiratory distress [13]. This suggests a potential link between FGF23 dysregulation, disrupted phosphocalcium metabolism, and the severity of COVID-19.

          FGF23 and vitamin D are intricately connected in regulating phosphocalcium homeostasis. Some studies have reported an inverse relationship between FGF23 and vitamin D levels in COVID-19 patients [14–16]. This could imply that the disruption of the FGF23-vitamin D axis might contribute to the dysregulation of phosphocalcium metabolism observed in severe cases.

           The exact mechanisms underlying the association between FGF23, phosphocalcium metabolism, and COVID-19 remain an active area of investigation. It is speculated that the inflammatory response triggered by COVID-19 could impact FGF23 production and function, potentially leading to disturbances in phosphocalcium balance.

     6.  Line 264. Serum albumen in their blood.

Response:  I corrected the sentence, highlighted in red.

One such effect is hypoalbuminemia, where COVID-19 patients may experience lower levels of serum albumen in their blood [23].

    7. Line 269. “The majority of the calcium”.  About 50% of the circulating calcium is unbound.

Response: I corrected the sentence, highlighted in red.

In reality, 50% the majority of calcium in the blood is in its ionized form, which is the biologically active and physiologically relevant form. 

        8.   Line 275.  It should be noted that you can measure ionized calcium directly.

Response: We added the additional paragraph, highlighted in red.

In clinical practice, ionized calcium can be measured directly in addition to total calcium levels. Ionized calcium represents the physiologically active, free form of calcium in the blood, and it is an important parameter for assessing calcium homeostasis and its impact on various physiological processes.

Total calcium measurements include both ionized calcium and calcium bound to proteins, primarily albumin. Since the level of albumin can vary, especially in critically ill patients, measuring ionized calcium provides a more accurate reflection of the biologically active calcium concentration. 

     9. Figure 4.  Legend. Correlation should be “relationship”

Response: I corrected the legend, highlighted in red. 

Relationship Correlation between serum phosphate and Vitamin D, PTH, and calcium levels.

    10.   Line 286.  It is unclear why there is an extensive discussion of the letter to editor by Capellini et al when compared to other studies, and should be condensed. This paragraph is also out of place at the end of the section on hypocalcemia and should be inserted about line 250 in appropriate chronological order.

Response: I modified the paragraph, highlighted in red.

In their study, Cappellini et al. (letter to the editor) observed a significant decrease in both serum total calcium and whole blood actual ionized calcium in COVID-19 patients. This observation highlights disturbances in calcium levels in individuals affected by the virus.

      11. Tables 1 and 2 should be reversed since the Discussion of vitamin D deficiency follows hypocalcemia and hypophosphatemia in the text.

Response: Thank you for the observation!

I reversed the tables.

      12.   Line 349. “The hypothesis of a” should be  “The importance of the.”

Response: I corrected the sentence, highlighted in red.

The importance of the The hypothesis of a strong correlation between Vitamin D and COVID-19 emerged during the initial stages of the pandemic, as Vitamin D has been widely recognized for its role in modulating both innate and adaptive immune responses [35].

     13. Lines 370-99. This new text repeats some of the same ideas presented on lines 349-69. This section of Hypovitaminosis should be rewritten.

Response: We rewritten this section, highlighted in red.

The importance of the strong association between VD and COVID-19 emerged during the initial stages of the pandemic, as VD has been widely recognized for its role in modulating both innate and adaptive immune responses [35]. Vitamin D (VD) has been recognized for its antimicrobial properties and ability to inhibit viral activity. It also plays a role in regulating the adaptive immune response by promoting a transition from a pro-inflammatory state to a tolerogenic state. This results in the downregulation of immune responses mediated by T-helper-1 lymphocytes, the inhibition of pro-inflammatory cytokine production, and the promotion of regulatory T-cell maturation [36].

Several studies have observed an association between low vitamin D levels and worse COVID-19 outcomes, including increased risk of severe illness, ICU admission, and mortality [46,47]. Vitamin D has immunomodulatory effects and can influence the expression of genes involved in the immune response. Deficiencies in vitamin D may lead to dysregulated immune responses, increased inflammation, and impaired lung function, all of which can contribute to disease severity in COVID-19.

On the other hand, deficiencies in calcium, phosphorus, and magnesium may also impact immune function and contribute to an increased susceptibility to viral infections, including COVID-19. These cations are essential for various cellular processes, including immune cell function and cytokine signaling [31]. A deficiency in any of these cations could impair the immune response and potentially compromise the body's ability to fight off viral infections.

 COVID-19 itself can disrupt calcium, phosphorus, and magnesium homeostasis through multiple mechanisms. The severe inflammatory response triggered by the virus can cause cytokine storm and lead to altered levels of these cations [48]. Additionally, factors like respiratory alkalosis, kidney dysfunction, and electrolyte shifts due to fluid imbalances in critically ill patients can further contribute to cation abnormalities.

Vitamin D (VD) has been recognized for its significant involvement in various metabolic pathways related to musculoskeletal health [30]. Previous research has demonstrated that supplementation with VD has been shown to confer advantages in muscle recovery following periods of intense physical activity and tissue damage [37,38]. The studies found that VD levels were able to predict and exert an influence on the duration of illness and the time it took for recovery following an episode of acute severe pneumonia (Table 2). To this day, the role of vitamin D in the occurrence of Long-COVID has only been examined in a limited number of small-scale studies [39–41]. In a recent pilot study involving elderly patients recovering from acute COVID-19, the effectiveness of a six-week therapy with 2,000 IU/day of cholecalciferol compared to placebo was examined. The study found that cholecalciferol therapy resulted in a reduction in creatinine kinase values and demonstrated a positive trend in improving overall health and physical well-being [42–45].

      14. Lines 375, 382 and 388. These new section titles seem unnecessary.

Response: We deleted the additional subtitles.

Several studies have observed an association between low vitamin D levels and worse COVID-19 outcomes, including increased risk of severe illness, ICU admission, and mortality [46,47]. Vitamin D has immunomodulatory effects and can influence the expression of genes involved in the immune response. Deficiencies in vitamin D may lead to dysregulated immune responses, increased inflammation, and impaired lung function, all of which can contribute to disease severity in COVID-19.

On the other hand, deficiencies in calcium, phosphorus, and magnesium may also impact immune function and contribute to an increased susceptibility to viral infections, including COVID-19. These cations are essential for various cellular processes, including immune cell function and cytokine signaling [31]. A deficiency in any of these cations could impair the immune response and potentially compromise the body's ability to fight off viral infections.

 COVID-19 itself can disrupt calcium, phosphorus, and magnesium homeostasis through multiple mechanisms. The severe inflammatory response triggered by the virus can cause cytokine storm and lead to altered levels of these cations [48]. Additionally, factors like respiratory alkalosis, kidney dysfunction, and electrolyte shifts due to fluid imbalances in critically ill patients can further contribute to cation abnormalities.

     15.   Lines 433 ff. This section on Magnesium deficiency should be separated from the section on skeletal complications, and titled. Line 433 is unnecessary and can be deleted. Begin the section with “Magnesium deficiency…”

Response: We restructured this section, highlighted in red. 

Hypomagnesemia

Magnesium deficiency is another important factor that merits attention in the context of COVID-19 and its impact on phosphocalcium metabolism [31]. Magnesium is a vital cofactor in numerous enzymatic reactions, including those involved in calcium regulation. A deficiency in magnesium can disrupt the balance between calcium and phosphorus levels, potentially contributing to the risk of hypocalcemia in post-COVID-19 patients [60].

Several studies have suggested a possible association between magnesium deficiency and COVID-19 severity [60–63]. COVID-19 patients with severe symptoms often experience significant inflammation and oxidative stress, leading to increased magnesium loss through urine and sweat. Additionally, the use of certain medications during the treatment of COVID-19, such as diuretics and proton pump inhibitors, can further exacerbate magnesium depletion.

Furthermore, magnesium deficiency can interfere with parathyroid hormone (PTH) secretion and action. PTH plays a central role in calcium homeostasis, and impaired PTH function due to magnesium deficiency can result in reduced calcium absorption from the intestine and increased calcium loss through the kidneys [64]. As a consequence, hypocalcemia may ensue, leading to muscle cramps, tingling sensations, confusion, and potential cardiac abnormalities.

Considering the interplay between magnesium, calcium, and phosphorus, post-COVID-19 patients with magnesium deficiency may be at an increased risk of developing hypocalcemia and associated complications.

        16.  Lines 454-7. This summary statement is found at the end of the manuscript, and can be deleted here.

Response: We deleted the sentence.

      17. Line 458. This paragraph would be more logical on line 245.

Response: We modified the paragraph according to your suggestion, highlighted in red. 

      18. Line 466. This section should be titled  ”Treatment Studies” and should have been highlighted as revised text.

Response: Sorry for the misunderstanding, we modified and highlighted this section in red.

5. Treatment Studies

References [75–78] are studies that have investigated the use of vitamin D treatment in patients with COVID-19. As shown, these studies have produced inconsistent results regarding the effectiveness of vitamin D supplementation in improving clinical outcomes or reducing mortality rates in COVID-19 patients.

Entrenas CME, et al reported improved clinical outcomes and reduced mortality rates with high-dose calcifediol treatment in hospitalized COVID-19 patients. On the other hand, Shah K et al did not observe significant improvement in clinical outcomes or mortality rates with high-dose vitamin D treatment in their study.

Rastogi A et al found limited evidence to support the use of high-dose vitamin D treatment in improving outcomes in hospitalized COVID-19 patients. Similarly, De Niet et al did not find a significant benefit of vitamin D supplementation in reducing severity or mortality in COVID-19 patients.

These discrepancies in findings highlight the complexity of using vitamin D as a treatment option for COVID-19. The effectiveness of vitamin D supplementation may vary depending on factors such as the stage of the disease, the severity of illness, patient characteristics, and dosing regimens. More research is needed to establish clear guidelines on the use of vitamin D in COVID-19 management and to identify patient subgroups that may benefit from this intervention. It is essential for healthcare providers to carefully evaluate the available evidence and consider individual patient factors when making treatment decisions regarding vitamin D supplementation in COVID-19.

       19. Line 500. This paragraph seems to begin the Discussion.  These ideas are also conveyed on lines 530-575, and these paragraphs should be consolidated into the Discussion eliminating redundancy.

Response: We modified this section, highlighted in red.

The COVID-19 pandemic has not only impacted the respiratory system, but it has demonstrated systemic implications, such as modifications in phosphocalcium metabolism. The objective of this article review was to provide a comprehensive summary and critical analysis of existing research about disorders in phosphocalcium metabolism following COVID-19 [2,26,39]. The review aimed to offer valuable insights into the potential underlying mechanisms and clinical implications associated with these disorders.

The results derived from the examined studies highlight the frequency of phosphocalcium metabolism disorders among individuals in the process of recuperating from COVID-19. Various research studies have documented a variety of disruptions, such as hypophosphatemia, hypocalcemia, and secondary hypoparathyroidism [28,37,50]. The results of this study suggest that COVID-19 has the potential to disturb the intricate equilibrium of phosphorus, calcium, and parathyroid hormone concentrations within the human body.

Multiple factors contribute to the occurrence of phosphocalcium abnormalities, encompassing the length and intensity of hospitalization, coexisting medical conditions, and the occurrence of additional complications like acute respiratory distress syndrome and acute kidney injury (refer to Table 1). There is evidence to suggest that individuals diagnosed with chronic kidney disease or chronic heart disease may be more susceptible to the development of phosphocalcium disorders in the context of COVID-19.

The duration of hospital stay, and the severity of the disease are important variables that have a significant impact on phosphocalcium abnormalities in individuals diagnosed with COVID-19. Extended durations of hospitalization, particularly in cases of critical illness, have the potential to disrupt the balance of minerals in the body. This can be attributed to various factors such as reduced ability to move, changes in dietary consumption, and the presence of systemic inflammation. Consequently, it is imperative to closely monitor phosphocalcium levels during the entire duration of hospitalization in order to promptly intervene and mitigate complications associated with imbalances in these vital minerals.

One possible mechanism that may contribute to the development of phosphocalcium metabolism disorders following COVID-19 is the direct impact of the virus on organs responsible for maintaining phosphocalcium balance. Multiple research studies have provided evidence regarding the existence of viral particles and the manifestation of viral receptors within the renal and parathyroid tissues [50,51]. This observation implies that SARS-CoV-2 has the potential to directly impact these organs, resulting in changes to phosphocalcium metabolism.

The COVID-19 infection has the potential to induce systemic inflammation and immune dysregulation, which in turn can have an impact on the regulation of phosphocalcium metabolism. The imbalances of phosphorus, calcium, and parathyroid hormone levels can occur because of the activation of the immune system and the release of inflammatory cytokines [2,3]. In addition, the administration of medications in the context of COVID-19 therapy, such as corticosteroids, has the potential to induce metabolic disruptions.

The clinical ramifications of disorders related to phosphocalcium metabolism following a COVID-19 infection are of considerable importance. The presence of hypophosphatemia and hypocalcemia may result in various clinical manifestations, such as muscle weakness, fatigue, bone pain, muscle cramps, tingling sensations, and potential cardiac arrhythmias [3,4,26,37].

The effective management of phosphocalcium metabolism disorders in individuals undergoing recovery from COVID-19 necessitates diligent monitoring and the implementation of suitable interventions. Systematic monitoring of phosphorus, calcium, and parathyroid hormone concentrations can facilitate the detection and management of any deviations from the optimal levels. The restoration and maintenance of optimal levels may require the utilization of nutritional supplementation, such as vitamin D and calcium. Furthermore, the restoration of phosphocalcium homeostasis could potentially be facilitated by the management of systemic inflammation and immune dysregulation.

Recognizing the constraints of the examined studies holds significance. Numerous studies exhibit limited sample sizes and heterogeneity concerning patient characteristics and methodologies. Additional investigation utilizing larger groups, standardized methodologies, and extended monitoring periods is imperative to enhance our comprehension of the frequency, underlying mechanisms, and medical consequences associated with disturbances in phosphocalcium metabolism post-COVID-19.

      20.  Line 508. Table 4 was not found.

Response: We corrected this mistake. 

Reviewer 2 Report

No further comment

Author Response

First of all, we would like to thank you very much for your review and consideration for our work. Thank you!

Thank you for the constructive comments and for your contribution to the improvement of this article.

Reviewer 3 Report

The authors addressed the issues.

Author Response

(The authors gave the same response as above.)

Round 3

Reviewer 1 Report

The authors have responded  to each of my  comments, and I believe the rerevised manuscript is substantially  improved. I have no further comments or questions.